# Antimicrobial Peptides Originating from Expression Libraries of *Aurelia aurita* and *Mnemiopsis leidyi* Prevent Biofilm Formation of Opportunistic Pathogens

**DOI:** 10.3390/microorganisms11092184

**Published:** 2023-08-29

**Authors:** Lisa Ladewig, Leon Gloy, Daniela Langfeldt, Nicole Pinnow, Nancy Weiland-Bräuer, Ruth A. Schmitz

**Affiliations:** 1General Microbiology, Kiel University, Am Botanischen Garten 1-9, 24118 Kiel, Germany; 2Institute of Clinical Molecular Biology (IKMB), Kiel University, Am Botanischen Garten 11, 24118 Kiel, Germany

**Keywords:** biofilm, antimicrobial peptide, AMP, pathogen

## Abstract

The demand for novel antimicrobial compounds is rapidly growing due to the rising appearance of antibiotic resistance in bacteria; accordingly, alternative approaches are urgently needed. Antimicrobial peptides (AMPs) are promising, since they are a naturally occurring part of the innate immune system and display remarkable broad-spectrum activity and high selectivity against various microbes. Marine invertebrates are a primary resource of natural AMPs. Consequently, cDNA expression (EST) libraries from the Cnidarian moon jellyfish *Aurelia aurita* and the Ctenophore comb jelly *Mnemiopsis leidyi* were constructed in *Escherichia coli*. Cell-free size-fractionated cell extracts (<3 kDa) of the two libraries (each with 29,952 clones) were consecutively screened for peptides preventing the biofilm formation of opportunistic pathogens using the crystal violet assay. The 3 kDa fraction of ten individual clones demonstrated promising biofilm-preventing activities against *Klebsiella oxytoca* and *Staphylococcus epidermidis*. Sequencing the respective activity-conferring inserts allowed for the identification of small ORFs encoding peptides (10–22 aa), which were subsequently chemically synthesized to validate their inhibitory potential. Although the peptides are likely artificial products from a random translation of EST inserts, the biofilm-preventing effects against *K. oxytoca*, *Pseudomonas aeruginosa*, *S. epidermidis*, and *S. aureus* were verified for five synthetic peptides in a concentration-dependent manner, with peptide BiP_Aa_5 showing the strongest effects. The impact of BiP_Aa_2, BiP_Aa_5, and BiP_Aa_6 on the dynamic biofilm formation of *K. oxytoca* was further validated in microfluidic flow cells, demonstrating a significant reduction in biofilm thickness and volume by BiP_Aa_2 and BiP_Aa_5. Overall, the structural characteristics of the marine invertebrate-derived AMPs, their physicochemical properties, and their promising antibiofilm effects highlight them as attractive candidates for discovering new antimicrobials.

## 1. Introduction

Multicellular organisms evolved in the presence of microbes, which play fundamental roles in their health, development, and evolution [1]. Early-branching metazoans, i.e., invertebrates, have required complex systems for recognizing and discriminating beneficial and pathogenic microorganisms [2,3,4,5]. However, unlike vertebrates, invertebrates lack classical antibody-based adaptive immunity [2]. Invertebrate immunity originates from defense reactions (i.e., innate immune system), including hemocyte-mediated modules, encapsulation, phagocytosis, and generation of antimicrobial peptides (AMPs) [6]. AMPs are small, mainly positively charged peptides that promote the innate defense mechanism by targeting the negatively charged membranes of microorganisms [7]. AMPs become embedded in the hydrophobic regions of lipid membranes, often forming pores, thus destabilizing biological membranes and causing cell lysis [8,9]. AMPs can also exhibit interactions with intracellular targets, contributing to their multifaceted antimicrobial activity [10]. AMPs have been shown to traverse the cell membrane and access intracellular compartments, where they can interact with various intracellular molecules and processes. These interactions can disrupt intracellular components, such as DNA, RNA, proteins, and enzymes, affecting vital cellular functions [10]. Such interactions may further enhance the bactericidal or microbicidal effects of AMPs and contribute to their overall antimicrobial potency [11]. AMPs are rapidly induced in response to microbes to modulate the immunoreactions. They have a wide range of inhibitory effects against bacteria, fungi, parasites, and viruses [12]. Consequently, AMPs have awakened interest as potential next-generation antibiotics, since emerging antibiotic resistance in pathogenic bacteria is a serious challenge and has led to the need for new alternative bioactive molecules less prone to bacterial resistance [13,14]. One particular interest in using AMPs is to combat biofilms of pathogens that have been demonstrated to be significant contributors to diseases [15,16]. Multiple factors contribute to the overall resistance of biofilms against antibiotics, including reduced metabolic and growth rates, protection by extracellular polymeric substances, and specific resistance mechanisms conferred by the altered physiology of biofilm bacteria [17,18]. Focusing on treating the highly challenging, adverse effects of harmful biofilm-associated infections, AMPs are indicated to have strong potential as antimicrobials and antibiofilm agents [15,19,20,21]. Marine invertebrates have been shown to be a primary source of natural AMPs [6] and are, therefore, likely to be an excellent source of novel marine AMPs. Over the past decades, several AMPs have been identified and isolated from marine invertebrates, including mollusks (defensins, mytilins, and myticins) [22,23,24], annelids (lumbricins and arenicins) [25], arthropods (cecropins and attacins) [26], and tunicates (styelins and didemnins) [27,28,29] but also cnidarians [30]. Cnidarians produce a range of AMPs with different structures and functions, such as ShK-1 from the sea anemone *Stichodactyla helianthus* [31] and Alyteserin-2a from the sea anemone *Anemonia sulcate* [32]. Cathelicidin-like peptides have been identified in several cnidarian species, including the sea anemones *Nematostella vectensis* and *Aiptasia pallida* [33]. Ctenophores are so far not listed within the existing antimicrobial-specific databases, such as InverPep (https://ciencias.medellin.unal.edu.co/gruposdeinvestigacion/prospeccionydisenobiomoleculas/InverPep/public/home_en, accessed on 15 January 2023). Invertebrate peptides have been shown to have antimicrobial activity against bacteria, fungi, and some viruses, and a couple may also have antitumor and immunomodulatory effects. The diversity and complexity of AMPs produced by marine invertebrates suggest that these compounds may have broad applications in medicine and biotechnology and provide insights into the evolution of host defense mechanisms. Those peptides possess novel and unique structures by only exhibiting a few side effects [6]. However, more research is necessary to fully understand the properties and potential applications of AMPs [34]. Understanding the mechanism of action of AMPs is critical to their development as therapeutic agents [35]. The optimal conditions for the activity of AMPs, their target sites within microbial cells, and how they interact with the host immune system have to be studied. AMPs may have toxic effects on the host, and there is a need to determine the appropriate dosages and potential side effects [36]. Further, the efficacy of AMPs may be affected by factors such as pH, ionic strength, and other environmental conditions. Therefore, research is required to optimize the delivery of AMPs to their target sites. Research should further focus on understanding the mechanisms of potential resistance development [37]. But, first and foremost, new AMPs with potent antimicrobial activity against a broad range of microorganisms have to be identified, which can be facilitated by using high-throughput screening approaches and bioinformatics [38,39].

The present study aimed to identify novel antimicrobial peptides from lower metazoans as promising and prosperous sources of novel biologically active compounds to prevent biofilms. Consequently, cDNA expression (EST) libraries were constructed from the Cnidarian moon jellyfish *Aurelia aurita* and the Ctenophore comb jelly *Mnemiopsis leidyi*. Low-molecular-weight fractions (<3 kDa) derived from both EST libraries were functionally screened to identify (artificial) activities preventing the biofilm formation of opportunistic pathogens. After identifying the sORFs in the respective inserts, the corresponding peptides were chemically synthesized and tested for their potential to prevent static and dynamic pathogenic biofilms.

## 2. Materials and Methods

### 2.1. Aurelia aurita Polyp Husbandry

The husbandry of polyps is described in detail in previous studies by Weiland-Bräuer et al. [40,41]. Polyps of the subpopulation North Atlantic (Roscoff, France) were kept in the laboratory at 20 °C in 30 PSU artificial seawater (Tropical Sea Salts, Tropic Marin) and fed twice a week with freshly hatched *Artemia salina* (HOBBY, Grafschaft-Gelsdorf, Germany).

### 2.2. Sampling of Aurelia aurita Polyps and Mnemiopsis leidyi Medusae

*A. aurita* polyps were used from husbandry by removing individual polyps with a disposable pipette. Individual *M. leidyi* medusae (with a mean umbrella diameter of 4 cm) were sampled from the Kiel Bight, Baltic Sea (54°32.8′ N, 10°14.7′ E) in May 2017 using a dip net. The animals were immediately transported to the laboratory and washed thoroughly with sterile artificial seawater.

### 2.3. Direct mRNA Isolation and Construction of the cDNA Expression Libraries

The mRNA of the *A. aurita* polyps and *M. leidyi* medusae were isolated with the DynaBeads^®^ mRNA DIRECT Micro Kit (Ambion, Austin, TX, USA) according to the manufacturer’s protocol “mRNA isolation from tissues”. In total, 28 parallel preparations were conducted using pools of 10 *A. aurita* polyps and 1 × 1 cm parts of *M. leidyi* medusae (in total 10 medusae) per isolation. Animal tissues were frozen in liquid nitrogen and homogenized with a motorized pestle (polyps) or a blender (medusae). For *A. aurita* 3 µg and for *M. leidyi* 1.2 µg mRNA were used to construct the cDNA expression libraries with the Clone MinerII cDNA Library Construction Kit (Invitrogen, Waltham, MA, USA) according to the manufacturer’s protocol. First, the mRNA was transcribed into cDNA and, subsequently, cloned into the pDONR222 entry vector. Second, the entry clones were recombined with Gateway expression vector pET300/NT-DEST using the ChampionTM pET300/NT-DEST GatewayTM Vector Kit (Invitrogen, Waltham, MA, USA) to create expression clones. The resulting plasmids retained the original alignment and reading frame, allowing for the functional analysis of full-length genes and entire libraries. SoluBL21 electrocompetent *Escherichia coli* cells were used for the electroporation and as background strain for the expression (Genlatis, San Diego, CA, USA). Each library consisted of 29,952 single clones and were stored in 96-well microtiter plates at −80 °C in the presence of 8% DMSO as cryoprotectant.

### 2.4. Preparation of Cell-Free Size-Fractionated Cell Extracts

The cDNA clones were first tested as 96 clone pools to identify antimicrobial peptides. Positive pools were successively screened as 48 and 24 clone pools and, finally, as single clones to efficiently screen the two cDNA libraries, each consisting of over 29,000 clones, in a high throughput procedure (see ref. [42]). Therefore, single cDNA clones were grown in 200 µL Luria Bertani medium (LB, Carl Roth, Karlsruhe, Germany) supplemented with 100 µg/mL ampicillin overnight at 37 °C in 96-well microtiter plates. Initially, 96 clones were pooled. The mixture of bacterial cultures was centrifuged at 9000× *g* for 5 min, and the bacterial pellet was resuspended in 250 µL 50 mM Tris/HCl buffer (50 mM NaCl, pH 7.8). The suspension was transferred to screw-cap tubes (Sarstedt, Nümbrecht, Germany) containing one glass bead of a diameter 2.7 mm and approximately 50 mg of glass beads of a diameter 0.1 mm (Carl Roth, Karlsruhe, Germany). Snap-frozen (liquid N_2_) bacterial cells were mechanically disrupted with Precellys^®^ (Bertin instruments, Montignyle-Bretonneux, France). The cell extract was sterile-filtered through a 0.2 µm centrifugal filter (Amchro GmbH, Hattersheim am Main, Germany) followed by size-fractionation using a 3 kDa centrifugal filter according to the manufacturer’s instructions (Merck KGaA, Darmstadt, Germany). The samples were kept at 4 °C or on ice throughout the process. The fractions < 3 kDa were collected, and the approximate protein concentrations were measured using a NanoDrop1000 (Thermo Fisher Scientific, Waltham, MA, USA) and adjusted to the same concentration. The empty pET300/NT-DEST/*E. coli* SoluBL21 was used as a control. Biofilm-preventing 96-pools were further tested in pools of 48 and 24 clones to rapidly unravel single clone(s) responsible for the biofilm inhibition using the described procedure.

### 2.5. Bacterial Biofilm Prevention In Vitro Assay

The opportunistic pathogenic bacteria *K. oxytoca* (DSM 7342, M5a1-type strain), *P. aeruginosa* PAO1 (DSM 1707, pathogenic-type strain), *S. epidermidis* RP62A (DSM 28319, clinical isolate from catheter sepsis), and *S. aureus* (DSM 11823, clinical isolate) were grown in 5 mL LB medium overnight at below-indicated temperatures. The cell concentrations of the overnight cultures were analyzed with Neubauer cell counting and set to 3 × 10^8^ cells/mL using GC minimal medium (with 1% (*v*/*v*) glycerol and 0.3% (*w*/*v*) casamino acids) [43] for *K. oxytoca* and Caso bouillon (17 g/L casein peptone, 3 g/L soybean peptone, 5 g/L NaCl, 2.5 g/L K_2_HPO_4_, and 2.5 g/L glucose) for the remaining strains in the crystal violet assay. Cultures were aliquoted in 96-well plates (180/195 µL for each cavity). Cell-free size-fractionated cell extracts (20 µL) and synthesized peptides (various concentrations, 5 µL) were added to the cultures. MTPs were closed with gas-permeable membranes (Breathe-Easy^®^, Diversified Biotech, Dedham, MA, USA) and incubated at 37 °C (*P. aeruginosa*, *S. aureus*, and *S. epidermidis*) or 30 °C (*K. oxytoca*) for 18 h without shaking. Three biological replicates were performed, each with eight technical replicates. Medium controls were performed for normalization. Biofilm formation was monitored and quantified using the crystal violet assay. Following an 18 h incubation period, the cell cultures were removed, and the wells underwent two washes with H_2_O. Biofilms that developed and adhered to the surfaces were subjected to staining using 200 μL of a 0.1% crystal violet solution (Carl Roth, Karlsruhe, Germany) for 10 min at room temperature (RT). The stained biofilms were left to air dry after eliminating the crystal violet solution through two H_2_O washes. A total of 200 μL of 96% ethanol was introduced into each stained well to solubilize the dye, and incubation at RT for 15 min followed. Biofilm formation was assessed by measuring the absorbance of resolved crystal violet at 590 nm with the plate reader Spectra max Plus 384 (Molecular Devices, Ismaning; Germany) [44,45]. The screening procedure is depicted in Appendix A.

In addition, the potential growth-inhibiting effects of synthetic peptides were tested on planktonic-growing pathogens. Again, 3 × 10^8^ cells/mL diluted in LB medium were aliquoted in 96-well plates (200 µL for each cavity). Synthetic peptides were added to two selected final concentrations of 3.5 µg/mL and 112.5 µg/mL at the beginning of the experiment. MTPs were closed with gas-permeable membranes and gently shaken at 80 rpm and 37 °C (*P. aeruginosa*, *S. aureus*, and *S. epidermidis*) or 30 °C (*K. oxytoca*) for 18 h. A comparative endpoint determination (growth control, peptide control IDR-1018, and identified peptides) of the turbidity at 600 nm was measured with plate reader Spectra max Plus 384. An unpaired *t*-test was conducted as a statistical hypothesis test to determine the significance (*p*-value) of growth differences.

### 2.6. Plasmid Preparation, Insert Size Determination, and Sequencing

The plasmid DNA of the identified biofilm-preventing single clones was isolated from 5 mL overnight cultures using the Presto™ Mini Plasmid Kit, according to the manufacturer (Geneaid, Taiwan). A restriction digest with the *Bsr*GI enzyme was performed for insert size determination, followed by subsequent gel analysis. The plasmids were Sanger-sequenced by the Institute of Clinical Molecular Biology, CAU Kiel, Germany. Sequences were analyzed with Geneious Prime software (version 2022.2.1) (Biomatters, Auckland, New Zealand). Vector sequences and PolyA tails were removed from the raw sequences. Cleaned insert sequences were used for open reading frame identification. The peptide sequences within the frame of the vector coding histidine-tag were selected for peptide synthesis. All sequence data can be found in Appendix A.

### 2.7. Synthesis of Peptides

Peptides with sufficient biofilm-preventing activity and a length of >5 amino acids were synthesized using conventional solid-phase peptide synthesis with >94% purity at Genscript (Leiden, The Netherlands) (Table 1). The synthetic peptides were dissolved in sterile distilled water (Carl Roth, Karlsruhe, Germany) and stored at −80 °C as 100 µL aliquots of 10 mg/mL.

### 2.8. Effect of Biofilm-Preventing Peptides on Biofilm Formation of K. oxytoca M5aI in a Microfluidic Flow Cell

The microfluidic flow cell system comprises a polymethyl methacrylate corpus with a single channel (1 × 8 × 0.1 mm). Two inlets and one outlet exist. A borosilicate glass (24 × 60 × 0.17 mm; Carl Roth, Karlsruhe, Germany) was fixed with the adhesive Black-Seal silicone (Weicon, Münster, Germany). Microliter syringes (Innovative Labor System GmbH, Stützenbach, Germany) were connected via polytetrafluoroethylene (PTFE) tubes (0.3 mm × 0.6 mm) (Bola, Grünsfeld, Germany) with the in- and outlets and clamped into a syringe pump (Model: 220) (KD Scientific, Holliston, MA, USA). The channel was sterilized by rinsing 70% EtOH for 24 h at a flow rate of 20 µL/h. Afterwards, the remaining ethanol was washed out with sterile water at 20 μL/h for 1 h. Subsequently, the channel was equilibrated with GC medium at 20 μL/h for 2 h. Inoculation was conducted with a cell suspension of 1 × 10^9^ cells/mL, generated from an overnight culture of *K. oxytoca* M5aI. Initial cell adhesion was ensured by incubation at 30 °C for 1 h without any flow. Next, the channel was continuously rinsed at 15 μL/h with GC medium injected within the first inlet. The synthetic peptides IDR-1018 (biofilm-preventing control peptide, [47,48], BiP_Aa_2, BiP_Aa_5, and BiP_Aa_6 were diluted in GC medium to a concentration of 25 µg/mL and injected within the second inlet. Thus, the final peptide concentration was 12.5 µg/mL, corresponding to 10 ng peptide within the 0.8 µL channel. Biofilm formation was conducted at a flow rate of 15 µL/h for 24 h at 30 °C. The biofilm formation of *K. oxytoca* M5aI was further studied without adding peptides. Four biological replicates were conducted for each treatment.

Biofilms were stained with the fluorescent dye Syto9 (488 nm) (Life Technologie, Carlsbad, CA, USA) and analyzed using the LSM 700 confocal laser scanning microscope (Zeiss, Oberkochen, Germany). Therefore, the microfluidic flow cells were washed with GC medium at 3 µL/min for 15 min. The channel was then filled with a 1:1000 dilution of the fluorescent dye Syto9 and incubated in the dark at RT for 30 min. Dye residues were removed by rinsing GC medium at 3 μL/min for 15 min. Four image stacks were recorded per flow cell with optical sections of 0.9 µm per z-step. The digital image acquisition, three-dimensional reconstruction, and calculation of the biofilm parameters were performed with the software “Zen Black” (version 14.0.22.201) (Zeiss, Oberkochen, Germany) and the microscopy image analysis software “Imaris” (version 9.9.0) (Oxford Instruments, Abingdon, UK).

## 3. Results

The expression libraries of the two basal metazoans, *A. aurita* and *M. leidyi*, were constructed and screened for biofilm-preventing peptides against opportunistic pathogenic bacteria (*K. oxytoca*, *P. aeruginosa*, *S. epidermidis*, and *S. aureus)* using the crystal violet assay. Promising peptide candidates were characterized for their potential to prevent the dynamic biofilm formation of *K. oxytoca* in microfluidic cells.

### 3.1. Construction of cDNA Expression Libraries of Two Basal Metazoans

Expression libraries were constructed from mRNA derived from the Cnidarian moon jellyfish *A. aurita* and the Ctenophore comb jelly *M. leidyi* into the Gateway expression vector pET300/NT-DEST (Invitrogen). Each library consisted of 29,952 single clones (pET300/NT-DEST in *E. coli* Solu BL21). According to the manufacturer, the generated plasmids of the expression clones retain the original alignment and reading frame of the insert, allowing for the functional analysis of full-length genes. However, there is also the possibility that random translation of short non-natural ORFs occurs. Plasmids of randomly selected clones (96 clones per library) were characterized by restriction analysis, demonstrating an average insert size of approximately 1.4 kbps and 95–98% insertion efficiencies for both libraries. Analyzing the insert sequences of those randomly selected clones revealed that 92% of the reads aligned to the respective genomes [46,49,50,51]. This, however, does not necessarily allow for any statement concerning their actual function in the medusae [46,49,50,51].

### 3.2. Identification of Biofilm-Preventing Clones from the cDNA Expression Libraries

Both of the constructed cDNA libraries were used to identify biofilm-preventing peptides derived from the basal metazoan hosts. Cell-free cell extracts were successively prepared from pools of 96, 48, 24, and single clones of the cDNA expression library (Appendix A). Notably, cell extracts were size-fractionated using 3 kDa cutoff columns, and only the fractions < 3 kDa were analyzed using the crystal violet assay. This assay was initially performed with the Gram-negative opportunistic pathogen *K. oxytoca* and the Gram-positive pathogen *S. epidermidis*. In total, 36% of the 96 clone pools showed biofilm-preventing activity for at least one of the two tested pathogens. Successively, pools of decreasing clone numbers were screened until single clones were identified (Appendix A). Cell-free size-fractionated cell extracts of single clones were evaluated for their biofilm-preventing potential, including two additional opportunistic pathogens, *P. aeruginosa* and *S. aureus*. Biofilm biomasses were measured using the crystal violet assay, with the biofilm formation of pathogens without any added test substance set at 100%. Overall, ten biofilm-preventing single clones (six derived from *A. aurita*, and 4 from *M. leidyi*) were identified with varying impacts on the biofilm formation of the different pathogens (Table 2). For instance, clones Aa_112_4H and Aa_127_8E consistently showed biofilm-preventing effects on all pathogens tested (Table 2). Clones Aa_127_8A and Aa_127_8F appear to have exclusive effects on *K. oxytoca* and *S. epidermidis* biofilm formation (Table 2). Clone Aa_127_8H notably enhances the biofilm formation of *K. oxytoca*, but it reduces the biofilm formation of the other bacteria (Table 2).

Sequence analysis of the biofilm inhibition-conferring inserts identified the corresponding small open reading frames (sORFs) (Appendix A). Here, exclusively those sORFs N-terminally fused to the vector-derived histidine-tag and longer than five amino acids were used for further analysis since shorter sequences were less likely to exhibit significant bioactivity. Structural models of the five respective peptides were predicted with PEP-FOLD 3 (Figure 1). NCBI BLASTp identified those peptide sequences in *A. aurita* and *M. leidyi* genomes. However, nonsignificant E-values were reached because of the short input sequences.

### 3.3. Synthetic Peptides—Excluding Cytotoxic Effects on Planktonic Bacteria

The five identified peptides were chemically synthesized and designated as BiP_Aa_2, BiP_Aa_4, BiP_Aa_5, BiP_Aa_6, and BiP_Ml_3 (Table 1). The calculated molecular weights (MWs) ranged between 1.14 (BiP_Aa_2) and 2.58 kDa (BiP_Aa_6) and the isoelectric points (pIs) between 7.95 (BiP_Aa_4) and 12.18 (BiP_Aa_5) (Table 1), offering insights into the peptide sizes, behavior under different pH conditions, solubility, electrophoretic behavior, and potential interactions with other molecules. The aggregation propensity was calculated from −0.091 (BiP_Aa_4) to 0.561 (BiP_Ml_3) (Table 1), indicating the formation of the protein aggregates necessary for assessing the biopharmaceutical and biotechnological potential. As a positive control, peptide IDR-1018 was additionally synthesized. IDR-1018 is known for its immunomodulatory and antimicrobial properties. It has been shown to possess antimicrobial activity against a wide range of bacteria, including both planktonic- and biofilm-forming bacteria. IDR-1018 can inhibit bacterial growth by disrupting bacterial cell membranes, interfering with essential cellular processes, and modulating the host immune response [47,48]. Initially, the potential growth-inhibiting effects of all synthetic peptides on planktonic-growing pathogens were evaluated to exclude the peptides’ growth-inhibitory (i.e., toxic) effects and focus on antibiofilm effects. Two contrasting concentrations of 3.5 and 112.5 µg/mL were selected. None of the peptides affected the growth behavior of any pathogens at a low concentration [47,48]. According to original reports [47,48], control peptide IDR-1018 administered at high concentration significantly reduced the planktonic growth of all strains (Figure 2, see Appendix A for *t*-test statistics). Host-deduced synthetic peptides BiP_Aa_4 and BiP_Aa_6 did not significantly inhibit planktonic growth, even at 112.5 µg/mL; moreover, they mostly resulted in growth promotion. The presence of BiP_Aa_2, BiP_Aa_5, and BiP_Ml_3 led to some pathogen-specific growth effects. In more detail, all three synthetic peptides reduced the planktonic growth of *P. aeruginosa* and *S. aureus* by 1.5–5%, while *S. epidermidis* was affected only by BiP_Aa_5 (3%). However, all observed growth effects were minimal in each case, although mostly statistically significant (Figure 2 and Appendix A).

### 3.4. Synthetic Peptides—Verification of the Biofilm-Preventing Effects on Static Biofilms

The biofilm-preventing potential of all synthetic peptides was tested in various concentrations ranging from 0.4 µg/mL to 112.5 µg/mL, including the two selected concentrations for cytotoxic effects exclusion, against the four opportunistic pathogens using the crystal violet assay (Figure 3). The presence of the control peptide IDR-1018 reduced the biofilm formation of *K. oxytoca* at a concentration of 3.5 µg/mL on average by 30% and at the highest concentration (112.5 µg/mL) by 70%. However, no interference with the biofilm formation of the second Gram-negative pathogen, *P. aeruginosa*, was observed. For Gram-positive *S. epidermidis*, the biofilm formation was reduced in a range of 10 to 30%, almost regardless of the concentration, whereas *S. aureus* showed a concentration-dependent reduction in biofilm formation by up to 60%.

For all of the identified host-derived synthetic peptides, the effects on the biofilm formation of at least one of the pathogenic strains were monitored (Figure 3). Peptide BiP_Aa_2 showed a negligible impact on *P. aeruginosa* and *S. aureus* biofilm formation at all peptide concentrations. *K. oxytoca* exhibited no significant change in biofilm formation until 14.1 µg/mL, but an effect manifested from 28.1 µg/mL (up to 43% reduction). *S. epidermidis* displayed a 45% maximum reduction. Moreover, the effect of BiP_Aa_2 on *S. epidermidis* biofilms appeared to reverse with increasing concentrations (Figure 3). BiP_Aa_4 had a limited effect on *S. aureus, P. aeruginosa,* and *K. oxytoca* biofilms. Moderate inhibition (23–47%) appeared in *S. epidermidis*, showing deviation from concentration dependence (Figure 3). BiP_Aa_5 effectively inhibited biofilm formation for all pathogenic strains studied. At the highest peptide dose (112.5 µg/mL), biofilm reduction reached 47% for *P. aeruginosa*, 44% for *S. aureus*, and 54% for *K. oxytoca*. *S. epidermidis* exhibited a milder reduction at this peptide concentration than the other strains (22%) (Figure 3). For BiP_Aa_6, *S. epidermidis* biofilm maximally reduced by 43%. *K. oxytoca* showed increasing biofilm inhibition, reaching 26% at the highest administered concentration, while *S. aureus* showed a 6–23% inhibition. *P. aeruginosa* biofilm initially increased and then gradually decreased, approaching control levels (Figure 3). BiP_Ml_3 had no significant effect on *S. aureus*, *P. aeruginosa*, or *K. oxytoca* biofilms, with *S. epidermidis* showing growth increase at low peptide doses. However, 112.5 µg/mL peptide progressively inhibited *S. epidermidis* biofilm, culminating in a 38% reduction at the highest concentration (Figure 3). In general, the biofilm formation of *P. aeruginosa* was not significantly impacted, but slight interference by BiP_Aa_5 and BiP_Ml_3 was obtained at a high concentration. Depending on the peptide, *S. epidermidis* biofilm formation was reduced by 5 up to 45%. Notably, the prevention occurred almost regardless of the peptide concentration, which was also the case for the control peptide. For *S. aureus*, BiP_Aa_2, BiP_Aa_4, and BiP_Ml_3 could not interfere with biofilm formation, whereas the presence of BiP_Aa_5 revealed a reduction of 40% (Figure 3).

Relating the growth inhibition data of the planktonic cells (Figure 2) when considering biofilm inhibition (Figure 3), four discernible patterns can be deduced that are specific to the tested pathogen and peptide (only results for the 112.5 µg/mL peptide concentration were regarded). Growth-inhibiting effects on planktonic cells correspond to noteworthy biofilm inhibition (e.g., BiP_Aa_5 in *P. aeruginosa* and *S.* spp.) or, conversely, growth-promoting effects align with augmented biofilm formation (e.g., BiP_Aa_4 and BiP_Aa_6 in *P. aeruginosa*). Furthermore, growth-promoting effects stand in contrast to biofilm inhibitory effects (e.g., BiP_Aa_6 and BiP_Ml_3 in *S. epidermidis*). Finally, BiP_Aa_2, revealed growth-inhibiting effects on planktonic cells but elicited increased biofilm formation. Consequently, we cannot establish overarching patterns. The complexity of correlating these effects might arise from the distinct lifestyles and behaviors of planktonic and biofilm cells, making direct comparisons between their impacts challenging.

In conclusion, most of the host-derived synthetic peptides prevented biofilms of Gram-negative and Gram-positive pathogenic bacteria in a concentration-dependent manner, with BiP_Aa_5 demonstrating the most substantial effects on biofilm formation for all tested pathogens. This finding argues that the initially observed biofilm-preventing activities of the 3 kDa cell extract fractions are mainly based on the respective sequence-identified peptides, although other inhibitory biomolecules present in the 3 kDa fractions cannot be excluded entirely.

### 3.5. Synthetic Peptides—Biofilm-Preventing Effects on Dynamic K. oxytoca Biofilms

In the first attempt, microfluidic flow cells were constructed and established for the biofilm formation of our biofilm model organism and opportunistic pathogen *K. oxytoca* (Figure 4A). A comprehensive biofilm formation of *K. oxytoca* was reached with an initial cell concentration of 8 × 10^5^ cells/channel and a flow rate of 15 µL/h for 24 h at 30 °C. A compact biofilm with a wavy surface was formed with a mean biofilm thickness of 11 ± 6 µm (Figure 4B, left bar) and volume of 112 ± 62 µm^3^ (Figure 4B, right bar, medium control). In a second step, the biofilm formation of *K. oxytoca* was analyzed in the presence of the synthetic host-derived peptides BiP_Aa_2, BiP_Aa_5, and BiP_Aa_6 and the control peptide IDR-1018 (10 ng/channel) with four biological replicates, each with four technical replicates (Figure 4B). In contrast to the medium control, the control peptide IDR-1018 significantly prevented biofilm formation. Predominantly, microcolonies but no 3D structures were detected, resulting in a reduced maximum thickness of 6 ± 2 µm (*p* = 0.0036; Figure 4B, left bar) and a reduced volume of 32 ± 16 µm^3^ (*p* = 0.0046; Figure 4B, right bar) Using the two host-derived synthetic peptides, BiP_Aa_2 and BiP_Aa_5, strong biofilm-preventing effects were observed, confirming the findings for static *K. oxytoca* biofilms. Single cells attached to the surface and formed microcolonies of 8 ± 3 µm (*p* = 0.1349; Figure 4B, left bar), resulting in a volume for BiP_Aa_2 of 34 ± 12 µm^3^ (0.0046; Figure 4B, right bar). Similarly, BiP_Aa_5 formed 7 ± 2 thick biofilms (*p* = 0.0342; Figure 4B, left bar) with a volume of 51 ± 25 µm^3^ (*p* = 0.0309; Figure 4B, right bar). The smallest effect was observed for BiP_Aa_6. Here, the formation of macrocolonies was detected, which reached a thickness of up to 8 ± 1 µm (*p* = 0.0674; Figure 4B, left bar) and a volume of 64 ± 36 µm^3^ (*p* = 0.1443; Figure 4B, right bar). However, those macrocolonies did not show a compact structure like that observed in the medium control (Figure 4B). Consequently, all tested synthetic peptides showed inhibitory effects on the dynamic biofilm formation of *K. oxytoca* in flow cells leading to a reduced biofilm thickness and volume.

## 4. Discussion

We identified small peptides of 10–22 aa length, derived from the cDNA expression libraries of two basal marine invertebrates, *A. aurita* and *M. leidyi,* with biofilm-preventing activities (Figure 1, Figure 3 and Figure 4B). These marine invertebrates potentially express the peptides as a natural defense, a component of their innate immune system. Examples of such small, cationic, and amphipathic peptides are already described for Arthropoda, Mollusca, and Urochordata [30]. A prominent Cnidarian AMP is Aurelin, purified from the mesoglea of *A. aurita* [52]. It is a 40 aa antimicrobial peptide with a molecular mass of 43 kDa with no reported homology to any known antimicrobial peptides. Aurelin has activity against Gram-negative and Gram-positive bacteria because of its structural features of defensins and channel-blocking toxins. Aurelin is processed from an 84 aa pre-proaurelin containing a putative signal peptide [52,53].

In our study, we decided to use functional metagenomics methods instead of bioinformatics-based analysis. Functional metagenomics involves cloning and expressing DNA or transcribed mRNA (cDNA) fragments from environmental samples in a heterologous host, such as *E. coli*. Although sequence-based approaches can identify many potential AMPs quickly and efficiently, predicting the function of these peptides can be challenging. Functional metagenomics can provide direct evidence of antimicrobial activity, but it may also lead to the identification of artificial AMPs with no ecological function in the natural system. Based on our experimental approach to identifying new AMPs in this report, it has to be taken into account that newly identified peptides might also be derived from a larger ORF or protein. During sequence analysis of the cloned inserts, we selected the first small ORF in-frame with the coding sequences of the vector backbone. However, the sORFs often did not reflect the complete insert sequence and might, thus, be non-natural. Most of the small peptides were embedded in larger genes, often encoding for housekeeping genes, like actin and ribosomal proteins (Appendix A). Consequently, the sORFs might be artificially generated based on the experimental design, and the encoded peptides identified likely do not necessarily possess an ecological antimicrobial activity in the natural system (jellies). On the other hand, because of the heterologous expression in *E. coli,* we potentially missed AMPs active against this opportunistic pathogenic bacterium.

Nevertheless, the biofilm-preventing activities of the five newly identified peptides were observed against *K. oxytoca*, *P. aeruginosa*, *S. epidermidis*, and *S. aureus* using the well-established crystal violet assay [44,45,54] (Figure 3). The crystal violet method’s weaknesses and limitations are well documented [39,55,56,57]. An inherent drawback is its indirect nature, offering no insight into cell viability or bacterial metabolic activity within biofilms. Reproducibility poses another concern, with biofilm detachment possible during the washing steps. Moreover, variances in the inoculum strength can yield disparate well-to-well outcomes, potentially causing false positives or negatives. Addressing this requires abundant replicates and strict adherence to the washing process throughout experimentation, as applied within the present study. Despite these limitations, the method’s cost-effectiveness and high sample throughput are merits [56].

The biofilm prevention was assayed parallel to the well-described 12 amino acids’ cationic AMP (VRLIVAVRIWRR), named IDR-1018 [19,47,48]. This innate defense component is derived from the natural bovine peptide Bactenecin and was reported to show antibiofilm activity against *P. aeruginosa*, *E. coli*, *Acinetobacter baumannii*, *K. pneumoniae*, Methicillin-resistant *S. aureus*, *Salmonella* Typhimurium, and *Burkholderia cenocepacia* [47,48]. The inhibitory activity of IDR-1018 is based on the induced dispersal of cells from biofilms at very low peptide concentrations (0.8 µg/mL). The underlying molecular mechanism was shown to involve the binding and subsequent degradation of stress-induced second messenger guanosine pentaphosphate ((p)ppGpp) [48]. In contrast, at higher concentrations (10 µg/mL), the peptide caused cell death of the biofilm cells [48]. In addition, a weak direct antimicrobial activity of IDR-1018 on planktonic cells was described (MIC 19 µg/mL) [58]. In this report, we verified an antimicrobial effect of IDR-1018 on planktonic bacteria at a high peptide concentration (112.5 µg/mL) (Figure 2), whereas, at a low concentration, no effect was revealed. Further, the reported 50% biofilm reduction for the pathogens *S. aureus* and *K. oxytoca* was reconfirmed in the here reported experiments. However, in contrast to the original study, *S. epidermidis* biofilm formation was only affected to a minor extent, and *P. aeruginosa* biofilms were almost unaffected.

Concentration-dependent biofilm-preventing properties were similarly observed for the jelly-derived peptides against all tested opportunistic pathogens, except *P. aeruginosa* (Figure 3). *P. aeruginosa* biofilm formation was only affected by BiP_Aa_5 and BiP_Ml_3 at a high concentration. A common concern around using AMPs as new antimicrobials is their high susceptibility to enzymatic degradation by proteases [12,38]. *P. aeruginosa* synthesizes many such proteases that are essential virulence factors to interfere with antibacterial defense mechanisms [59], which might be the reason for our observation. In previous reports, *P. aeruginosa* biofilm inhibition was often described as an antimicrobial effect on planktonic cells [60,61]. Our synthetic peptides BiP_Aa_2, BiP_Aa_5, and BiP_Ml_3 showed only low antimicrobial effects on planktonic bacteria (2–5% growth reduction). A notable exception to the concentration-dependent trend in biofilm prevention against the other opportunistic pathogens was observed for BiP_Aa_2, which revealed an increase in the biofilm quantity of *S. epidermidis* when the peptide concentration increased above 3.5 µg/mL. The preliminary results demonstrated that an increasing peptide concentration causes dimerization of BiP_Aa_2 (aggregation propensity: 0.201; Table 1), likely accompanied by a loss of activity, as shown for the ceratotoxin-like peptide from *Hypsiboas albopunctatus* [62,63]. Overall, peptide BiP_Aa_5 demonstrated the most substantial effects on biofilm formation for all tested pathogens. The reduction of biofilms was comparable to that of IDR-1018, resulting in a reduction of biofilm masses by up to 50%. This finding agrees with the high similarity of amino acid sequences of the two peptides. Further, peptides BiP_Aa_4 and BiP_Ml_3 were poorly soluble in hydrophilic solutions, thus potentially influencing the final peptide concentrations for the biofilm inhibition assays. Addressing this concern, a series of preliminary experiments evaluating commonly employed hydrophobic solvents, such as dimethyl sulfoxide (DMSO), acetonitrile, and ethyl acetate, were conducted before the study. Regrettably, these investigations revealed cytotoxic effects associated with all three solvents when exposed to at least one species of biofilm-forming bacteria. Notably, ethyl acetate led to a discernible loss of peptide activity. Thus, water was chosen as the solvent despite accepting the lower solubility of the two peptides BiP_Aa_4 and BiP_Ml_3.

The synthetic peptides, BiP_Aa_2, BiP_Aa_5, and BiP_Aa_6, showed the best effects on the Gram-negative model of biofilm formation of *K. oxytoca*. Consequently, their biofilm-preventing potentials were further validated and studied in spatial resolution in microfluidic flow cells with *K. oxytoca* (Figure 4). Microfluidics technologies can be applied to study bacterial adhesion and biofilm development by precisely controlling the fluidic environment and allows for imaging of the biofilm [64]. Microscale systems can address many of their macroscale counterparts’ disadvantages and measurement challenges [65]. Notably, such a system requires relatively fewer sample and lower reagent volumes and, thus, is advantageous in antibiofilm studies with antimicrobials. The biofilm formation of *K. oxytoca* was analyzed under the constant presence of synthetic peptides after one hour of initial cell adhesion. CLSM micrograph analyses of established biofilms after 24 h, in general, confirmed inhibitory impacts and showed reduced biofilm thickness and biofilm volume in the presence of all peptides tested (Figure 4). BiP_Aa_2 showed similar biofilm-preventing potential as the control IDR-1018 and, thus, even more prominent effects as observed for static biofilms. Imaging further enabled a comparison of the biofilm structures. *K. oxytoca* generally forms a compact, wavy structure, while adding the peptides revealed that only single cells attached to the surface, successively forming microcolonies. BiP_Aa_6 resulted in the formation of macrocolonies without a compact structure. These findings imply the disturbance of the biofilm matrix, interruption of bacterial cell signaling systems, and disruption or degradation of biofilm-embedded cells’ membrane potential, as reported for AMP actions against bacterial biofilms [66].

Overall, the five characterized invertebrate-derived peptides harbor the typical cationic structure with spatially separated hydrophobic and charged regions resulting in an amphipathic molecule that is probably crucial for their overall mode of action [67]. The molecular weights were calculated as <3 kDa (Table 1); thus, they were in the expected range through size exclusion, indicating typical binding interactions of small peptides with other molecules, such as receptors, enzymes, or proteins [7]. The isoelectric points of the peptides ranged between 8 and 12 (Table 1), reflecting that at a pH of 8–12, the peptides carry no net positive or negative charge. The pI of an AMP can influence its interactions with biological membranes, target cells, and other molecules. AMPs with positive charges are often attracted to negatively charged bacterial membranes, contributing to their antimicrobial activity [38]. Further, the aggregation propensity is associated with the formation of aggregates, which can lead to various issues, especially in biopharmaceutical and biotechnological applications. Primary structure, physicochemical properties, environmental conditions, concentration, and mechanical stress can influence the aggregation propensity of peptides [68]. A negative aggregation propensity value suggests that the molecule has a reduced tendency to aggregate, as shown for BiP_Aa_4 and BiP_Aa_6. It implies that the molecule is likelier to remain in its individual or monomeric state rather than form aggregates. A negative aggregation propensity can be favorable from a stability standpoint, as it indicates a lower risk of forming aggregates that could potentially lead to loss of function or other undesirable effects. In contrast, the remaining three identified peptides possess a positive aggregation propensity value, suggesting that the molecule has a higher tendency to aggregate, which could impact the molecule’s stability, solubility, and functionality. Peptides BiP_Aa_4 and BiP_Ml_3 showed a lower water solubility, which can be explained for BiP_Ml_3 by its aggregation propensity value of 0.561 (Table 1). Both increasing [69,70] as well as decreasing [63,71] antimicrobial properties have been described through peptide aggregation.

The majority of the identified peptides showed promising biofilm-preventing activities against selected Gram-negative and Gram-positive opportunistic pathogens and, thus, can serve as starting points to develop efficient and effective antibiofilm agents. Combating bacterial infections caused by biofilm-forming bacteria is a difficult task and a major challenge for healthcare systems and aquaculture [72,73]. AMPs have enormous potential as antibiofilm agents by harboring broad-spectrum antimicrobial activity, a low risk of fast-developing bacterial resistance, and can work synergistically with antibiotics [74]. However, there is still limited information on the interaction of AMPs with biofilm components [66]. More research is needed to understand their precise mechanisms of action on the molecular level, such as inhibiting quorum-sensing (QS) signals that control biofilm formation and interfere with signaling pathways involved in the synthesis of the biofilm matrix [72]. Although dozens of marine invertebrate-derived AMPs have been described, their (antibiofilm) mechanisms and applications are still little explored and provide an opportunity for comprehensive research on their use in disease treatment [6]. Our jelly-derived peptides, although likely not expressing a natural function in the medusa, already show promising broad-spectrum antibiofilm activities, and further studies on their mode of action, stability, and side effects can lead to their future application.

## Figures and Tables

**Figure 1 microorganisms-11-02184-f001:**
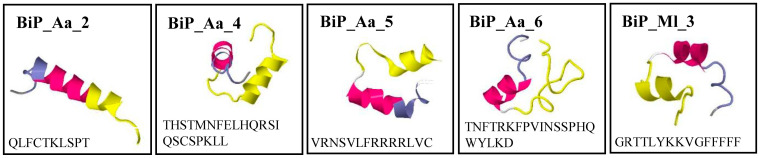
Predicted structural models of biofilm-preventing peptides. Sequences were analyzed using the Geneious Prime software (version 2022.2.1) (Biomatters, Auckland, New Zealand). Models were created using PEP-FOLD 3.

**Figure 2 microorganisms-11-02184-f002:**
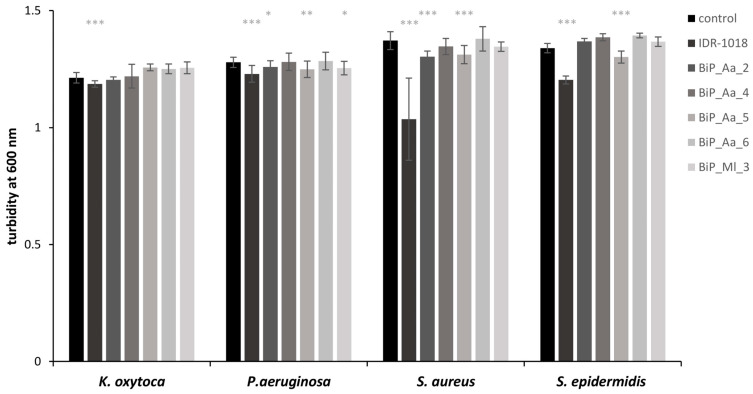
Exclusion of potential growth-inhibiting effects of the synthetic peptides on planktonic cells. *K. oxytoca*, *P. aeruginosa*, *S. epidermidis*, and *S. aureus* (3 × 10^8^ cells/mL) were grown for 18 h in 200 µL LB medium at 80 rpm after adding 112.5 µg/mL of the synthetic peptides. Turbidity was monitored at 600 nm of two biological replicates, each with eight technical replicates, represented as the means with corresponding standard deviations. The growth effects (promoting and inhibiting) effects were compared to a control without the addition of peptide. Significant *p*-values are indicated only for growth-inhibiting effects: * *p* < 0.05, ** *p* < 0.01, and *** *p* < 0.001.

**Figure 3 microorganisms-11-02184-f003:**
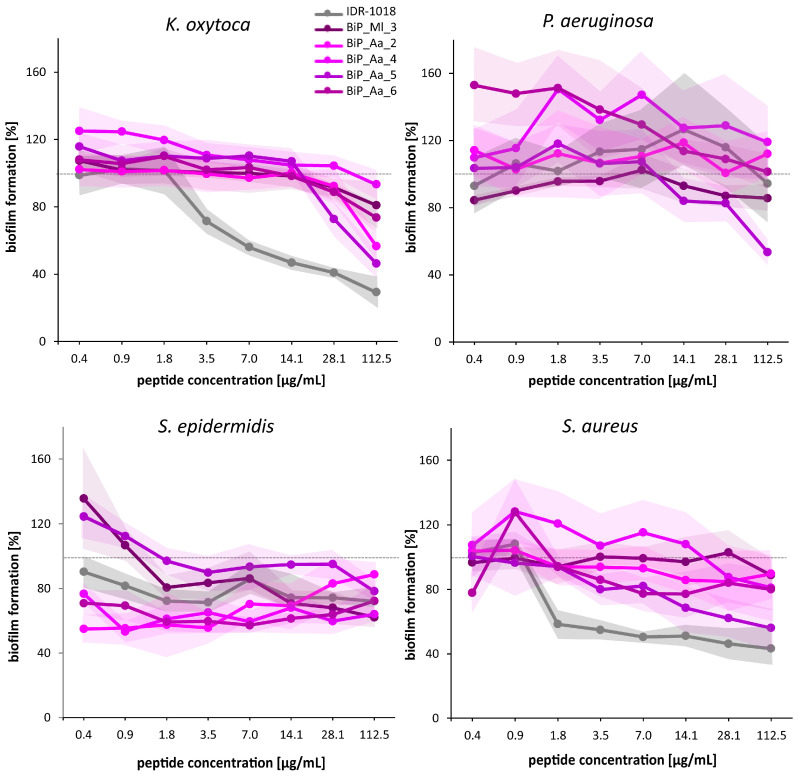
Biofilm-preventing effects of synthetic peptides on static pathogenic biofilms. Biofilm-forming strains *K. oxytoca*, *P. aeruginosa*, *S. epidermidis*, and *S. aureus* (3 × 10^8^ cells/mL) were grown for 18 h in 200 µL Caso Bouilion. Synthetic peptides were added from the beginning in concentrations of 0.4, 0.9, 1.8, 3.5, 7.0, 14.1, 28.1, and 112.5 µg/mL. Diagrams represent the average of three biological replicates, each with eight technical replicates. Biofilm formation was quantified using the crystal violet assay. The biofilm-preventing effect was calculated as a percentage value compared to the biofilm control (100%, dashed line); standard deviations are depicted as cloud envelopes.

**Figure 4 microorganisms-11-02184-f004:**
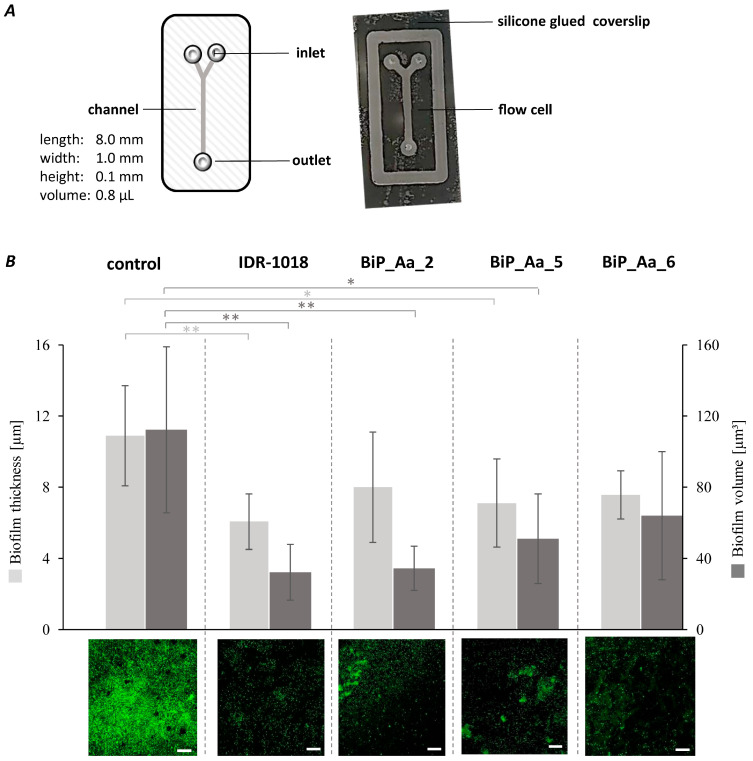
Biofilm prevention of *Klebsiella oxytoca* with selected synthetic peptides in microfluidic flow cells. (**A**) Microfluidic flow cells were constructed of polymethyl methacrylate with a single channel (8 × 1 × 0.1 mm). Silicone adhesive fixed the flow cell to a borosilicate coverslip (24 × 60 × 0.17 mm). (**B**) *K. oxytoca* biofilms were formed in GC minimal media for 24 h at 30 °C. Synthetic peptides were added in a concentration of 25 µg/mL (10 ng/channel) after 1 h of adhesion time. Peptides and medium were continuously supplemented at a flow rate of 15 µL/h for 24 h at 30 °C. Biofilms were stained with SYTO9 and analyzed with confocal laser scanning microscopy using the software Zen Black (version 14.0.22.201) and Imaris (version 9.9.0). The biofilm characteristics are presented in a bar plot as the means of four biological replicates, each with four technical replicates with the respective standard deviations (upper panel). Significance is represented by *p*-values: * *p* < 0.05 and ** *p* < 0.01. Representative CLSM images with scale bars representing 50 µm (lower panel).

**Table 1 microorganisms-11-02184-t001:** Synthetized peptides. Peptides were synthesized in 94% purity at Genscript (Leiden, The Netherlands). The characteristics of the peptides are summarized. MW_CALC_, calculated molecular weight; pI_CALC_, isoelectric point.

Peptide Designation	Related EST Clone	Amino Acid Sequence	MW_CALC_ (kDA)	pI_CALC_	Aggregation Propensity [46]
BiP_Aa_2	*A. aurita* 112_6C	QLFCTKLSPT	1.14	8.22	0.201
BiP_Aa_4	*A. aurita* 127_8E	THSTMNFELHQRSIQSCSPKLL	2.55	7.95	−0.091
BiP_Aa_5	*A. aurita* 127_8F	VRNSVLFRRRRLVC	1.78	12.18	0.150
BiP_Aa_6	*A. aurita* 127_8H	TNFTRKFPVINSSPHQWYLKD	2.58	9.7	−0.042
BiP_Ml_3	*M. leidyi* 010_9A	GRTTLYKKVGFFFFF	1.8	10.29	0.561
control IDR-1018 [47]		VRLIVAVRIWRR	1.5	12.48	0.487

**Table 2 microorganisms-11-02184-t002:** Identified single clones derived from the cDNA expression libraries of *A. aurita* and *M. leidyi* and their impact on the biofilm formation of pathogens. Biofilm biomasses were calculated based on the crystal violet assay. Biofilm formation of pathogens without adding a test substance was set as 100%.

Clone Designation	Biofilm Formation (%)			
	*K. oxytoca*	*P. aeruginosa*	*S. aureus*	*S. epidermidis*
Aa_112_4H	71 ± 12	85 ± 19	77 ± 9	55 ± 14
Aa_112_6C	92 ± 14	79 ± 24	84 ± 13	49 ± 9
Aa_127_8A	22 ± 2	100 ± 18	100 ± 13	77 ±13
Aa_127_8E	80 ± 3	69 ± 17	78 ± 13	6 ± 1
Aa_127_8F	28 ± 3	75 ± 18	84 ± 23	95 ± 10
Aa_127_8H	104 ± 5	77 ± 18	61 ± 10	40 ± 12
Ml_068_11H	107 ± 13	81 ± 13	69 ± 7	29 ± 3
Ml_011_11H	27 ± 2	100 ± 22	67 ± 3	77 ± 10
Ml_010_9A	71 ± 8	78 ± 14	76 ± 5	94 ± 9
Ml_010_9G	68 ± 14	75 ± 18	89 ± 17	91 ± 7

## Data Availability

All generated and analyzed data are part of the manuscript. A preprint of the manuscript is available at bioRxiv doi: https://doi.org/10.1101/2023.03.02.530746, accessed on 2 March 2023.

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
