# Peer review of "Antimicrobial Peptides Originating from Expression Libraries of Aurelia aurita and Mnemiopsis leidyi Prevent Biofilm Formation of Opportunistic Pathogens"

_microorganisms, 2023, doi:10.3390/microorganisms11092184_

Round 1

Reviewer 1 Report (New Reviewer)

Dear editors and reviewers

Thank you for submitting MS for review entitled”Antimicrobial peptides originating from expression libraries of  Aurelia aurita and Mnemiopsis leidyi prevent biofilm formation of opportunistic pathogens”

The MS is well written, having good ideas, aim and prestigious methodology. However, data presentation especially the main core of the MS representing the antimicrobial and the antibiofilm activity of the peptides is not clear and overestimated.

The MS lacking the figures and accession number of the expressed peptides.

The submitted tables are not discussed or even explained in the result of the discussion sections.

Materials and methods

-        In the Preparation of cell-free size-fractionated cell extracts

Why the authors prepared the protein extract from 96 pool first and then Biofilm-preventing 96-pools were further 144 tested in pools of 48 and 24 clones to unravel single clone(s) responsible for the biofilm 145 inhibition.

-        More details on the CV assay and assay of antimicrobials activity need to be added.

-        Why the authors sequenced only insert of a length of > 5 amino acids

-        How the authors calculated the effect on microbial growth? By viable count of measuring turbidity

-        On what bases the authors choose IDR-1018 as control.

-        How did the authors handle the poor solubility of some peptides, what was the solvent used, how did they detect the bacterial growth?

Result section

-        No figures are available for the MS.

-        Provide the accession numbers of the interests and the expressed protein.

-        Why the authors synthesized the five identified peptides an NOT express then from the purified clones.

-        Decreasing the planktonic growth of P. aeruginosa and S. aureus by 1.5 – 5 %, and S. epidermidis was by BiP_Aa_5 (3 %) does not represent and significant effect on the microbial growth. Why the authors did not study the effect of higher conc of the tested peptides on microbial growth.

-        The authors describe the effect of peptides on biofilm K. oxytoca biofilm formation was detected the most substantial effects, what does it mean what was the percentage reduction, is this represent a significant reduction.

-        Data description line 285-295 are unclear, detailed data explanation need to be added.

-        Why did the authors study and represent the inhibitory dynamic effect of peptides on K. oxytoca only what about the other tested isolates.

-        The authors did not retrieve or reveal to the submitted tables in the result or discussion sections.

Discussion

The authors discussed the activity IDR-1018 that used as control, in details (359-377) that are not the main core of the MS.

The MS is well written

Author Response

Reviewer 2 Report (New Reviewer)

Since antibiotic treatment of multidrug-resistant and biofilm-associated infections is very often ineffective, a search for new therapeutic solutions is necessary. In this context, your research is valuable from both scientific and practical point of view. The manuscript is really interesting and the research was well planned. However, I have some comments, which need to be addressed before the acceptance of the manuscript to be published in Microorganisms.

Major comments:

1.      The weaknesses of the research is still used but currently not recommended the crystal violet staining of biofilm. In future studies change the method for more precise and repeatable.

2.      Introduction (line 39-41): The term “cellular immunity” is confusing, since you mentioned both: cellular innate immune mechanisms (e.g. phagocytosis) and humoral innate immune components such as AMPs. It will be much better to say just: Invertebrate immunity.

3.      Introduction (line 42-45): Destabilizing cell membranes is in fact the main mechanism of AMPs’ action, but their also can interact with intracellular targets. Complete the information about this and provide appropriate references.

4.      Subsection 2.5 (line 149-150): Complete the information about K. oxytoca and S. aureus strains - whether they were reference strains, clinical isolates or your laboratory strains with unknown origin?

5.      Table 1 is completely redundant – growth conditions and culture media are similar and may be successfully described in the text (Materials and methods).

6.      Subsection 2.8 (line 191): I expect that it was about Fig. 4A

7.      Subsection 3.3 (line 264-277): In this fragment you described the results obtained for planktonic cultures, so it does not correspond to the title (“biofilm-preventing effects on static biofilms”). I suggest to prepare a separate subsection for studying the protein effect on planktonic cultures. Make it also clear in the description that the results for the lower concentration (3.5 µg/ml) are not shown. Complete information, which differences were statistically significant (give p values in brackets) and after description of the results (line 273-277) give the reference to appropriate Figure.

8.      Subsection 3.3 (line 295): “revealed a reduction of 40 %” – Since the percentage of reduction depended on BiP_Aa_5 concentration, improve the description giving the range of reduction.

9.      Subsection 3.3 (line 296-298): Based on obtained results a conclusion that “biofilms of Gram-negative and Gram-positive pathogenic bacteria were prevented in a concentration-dependent manner by most of the host-derived synthetic peptides” is highly exaggerated (there were almost no influence on P. aeruginosa biofilm and very weak effect on S. aureus biofilm). Improve the sentence.

Minor comments:

1.      Subsection 3.3 (line 271): Since the peptides were used only at one higher concentration (112.5 µg/ml) apply a singular noun; at concentration (not “in concentration”)

2.      Subsection 3.4: The title should be in Italics

3.      Subsection 3.4 (line 309-310): “a mean biofilm thickness of 11 ± 6 μm (Fig. 4B left bar) and volume of 112 ± 62 μm3 (Fig. 4B left panel, medium control)” – Unify description in reference to Fig. 4 not to be confusing, e.g. (thickness) left bar / (volume) right bar of control

4.      Discussion (line 364-365): Write E. coli in a short form. Correct spelling: Salmonella Typhimurium (second part without Italics and with capital letter, since this is serotype not the species name)

5.      Discussion (line 367 / 369): “very low peptide concentrations” / “higher concentrations” – change on: concentration (singular) since you gave only one value  

Round 2

Reviewer 1 Report (New Reviewer)

Dear authors/ editors

Thank you for submitting response to the comments on the MS " Antimicrobial peptides originating from expression libraries of Aurelia aurita and Mnemiopsis leidyi prevent biofilm formation of opportunistic pathogens" However, I still have concerns regarding the significance of the obtained data.

- Statistical analysis of figure 1 should be revised as the data seems to be non-significant.

- Figure 3, no statistical differences were attained with effect of the peptide on biofilm formation, compare the antibiotic activity of the tested peptides to the untreated planktonic cells with indication of significant difference at what concentration.

- Figure 3; Not %biofilm formation but % Biofilm inhibition

Dear authors/ editors

Thank you for submitting response to the comments on the MS " Antimicrobial peptides originating from expression libraries of Aurelia aurita and Mnemiopsis leidyi prevent biofilm formation of opportunistic pathogens" However, I still have concerns regarding the significance of the obtained data.

- Statistical analysis of figure 1 should be revised as the data seems to be non-significant.

- Figure 3, no statistical differences were attained with effect of the peptide on biofilm formation, compare the antibiotic activity of the tested peptides to the untreated planktonic cells with indication of significant difference at what concentration.

- Figure 3; Not %biofilm formation but % Biofilm inhibition

Author Response

This manuscript is a resubmission of an earlier submission. The following is a list of the peer review reports and author responses from that submission.

Round 1

Reviewer 1 Report

There's a novelty value in this work. I recommend its publication, but some minor revisions should be addressed before reconsideration. Several suggestions have been given below:

-       Language errors that require correction, e.g.:

L 132: Three biological were  performed, each with eight technical replicates... Should be rather: Three bioassays were…...

L346: which migth be..

L316: Consequently, some of the the sORFs might be artifically

L346: which migth be the reason for our obersvation.

L342: mass of S. epderimidis,

etc.

Author Response

Manuscript ID: microorganisms-2304942

Title: Antimicrobial peptides originating from expression libraries of Aurelia aurita and Mnemiopsis leidyi prevent biofilm formation of opportunistic pathogens

Authors: Lisa Ladewig, Leon Gloy, Daniela Langfeldt, Nicole Pinnow, Nancy Weiland-Bräuer, Ruth Anne Schmitz

Point-by-point response to Reviewer 1

We thank the reviewer for reviewing our manuscript and recommending publication after minor revision. The reviewer only recommended the correction of the following language errors:

L 132: Three biological were  performed, each with eight technical replicates... Should be rather: Three bioassays were…...

L346: …which migth be..

L316: Consequently, some of the the sORFs might be artifically

L346: which migth be the reason for our obersvation.

L342: mass of S. epderimidis, etc.

We apologize for the language errors. We carefully reviewed the manuscript and corrected all errors, supported by Grammarly Writing Assistant.

Reviewer 2 Report

The manuscript by Ladewig and colleagues reports the identification of putative novel AMPs from two abundant marine invertebrates, Aurelia aurita and Mnemiopsis leidyi.

The authors chose to use a somewhat old-fashioned, but potentially useful approach, based on cDNA cloning, followed by the identification of the clones expressing peptides with potentially interesting biological activity. Nevertheless, the use of this approach would have required the authors to actually understand what they were doing (in terms of the sequences they are cloning and understanding the limitations that this methodology), and to use extreme care in verifying the reliability of their predictions (using bioinformatics) prior to moving forward with the characterization of peptides that were likely to be artefacts. Since whole genome sequences are available for both species, the authors could have made a much better job in checking the reliability of the cloned sequences (regardless of what the manufacturer’s protocol states about the inserts being “in frame”), verifying whether the predicted peptides did actually match any known gene model. And, most importantly, verifying whether the full cloned cDNA sequence matched any  other protein in a different frame with respect with the frame that encoded the short peptide they inferred.

In detail, while the authors have demonstrated the biological activity of the isolated peptides, they have briefly discussed the possibility (at line 316) that “some of the the sORFs might be artificially generated based on the experimental design, and encoded peptides identified, do not necessarily possess an ecological antimicrobial activity in the natural system”.

This is an important acknowledgement, but the authors should most certainly have gone even further in the characterization of cloned sequences and, most importantly, they should have NOT assumed that inserts were necessarily in frame. As a matter of fact, some of the clones reported in table S1 are clearly not in frame, and therefore the inferred peptide sequence is not a real peptide (in its natural system). I would go even further and state that most likely NONE of the peptides identified and characterized in this study was a real peptide. A clear example is provided by clone Aa_127_8E, reported to encode a portion of a retinoid X receptor. However, the analysis of the nucleotide sequence reveals that this was an out-of-frame clone of the actin mRNA (which can be clearly spotted in frame 3). Ml_011_11H is also, quite clearly, an out-of-frame 60S ribosomal protein L13 cDNA. Ml_068_11H is an out-of-frame 40S ribosomal protein S25 cDNA.

Unfortunately, this invalidates, in my opinion, the entire work. Functionally characterizing (or even predicting the hypothetical structure) of non-existing peptides does not make much sense and, albeit some of the artificial peptides generated by random translation of short (non-existing) ORFs might indeed display significant biological activities, the fact itself that such sequences are not real is completely inconsistent the rationale of the whole study (using the two target organisms as a source of novel AMPs ).

Other comments.

Please check that scientific names are written in italics throughout the manuscript

L62: this is generally true, but I feel like this sentence is somewhat too optimistic, as only a fraction of all marine AMPs display a significant antimicrobial activity at concentrations that would make their biomedical application feasible, without showing at the same time no significant cytotoxicity. The development of AMP-based drugs for human use still has a long way to go.

L62: before the last paragraph of the introduction, I would suggest the authors to add a small additional paragraph to briefly summarize currently available information about ctenophore and cnidarian AMPs (i.e. no information whatsoever for the former, to the best of my knowledge, vs a few preliminary indications –including aurelin, among the others- for the latter. This would help the reader to understand that these organisms have been definitely under-exploited so far in terms of AMP research.

Materials and methods

One of the drawbacks of this approach is that some potentially interesting AMP sequences might have been missed due to the fact that they have a significant activity against E. coli, preventing the possibility to obtain viable bacteria expressing them. This should be somehow discussed at a later point in the manuscript.

The bottom part of figure 3 appears to have been cut

Author Response

Manuscript ID: microorganisms-2304942

Title: Antimicrobial peptides originating from expression libraries of Aurelia aurita and Mnemiopsis leidyi prevent biofilm formation of opportunistic pathogens

Authors: Lisa Ladewig, Leon Gloy, Daniela Langfeldt, Nicole Pinnow, Nancy Weiland-Bräuer, Ruth Anne Schmitz

Point-by-point response to Reviewer 2

We thank the reviewer for reviewing our manuscript. We are pleased to answer the questions and comments of the reviewer (in italics) in the following point-by-point response. Our answers are given in regular font.

The manuscript by Ladewig and colleagues reports the identification of putative novel AMPs from two abundant marine invertebrates, Aurelia aurita and Mnemiopsis leidyi.

The authors chose to use a somewhat old-fashioned, but potentially useful approach, based on cDNA cloning, followed by the identification of the clones expressing peptides with potentially interesting biological activity. Nevertheless, the use of this approach would have required the authors to actually understand what they were doing (in terms of the sequences they are cloning and understanding the limitations that this methodology), and to use extreme care in verifying the reliability of their predictions (using bioinformatics) prior to moving forward with the characterization of peptides that were likely to be artefacts. Since whole genome sequences are available for both species, the authors could have made a much better job in checking the reliability of the cloned sequences (regardless of what the manufacturer's protocol states about the inserts being "in frame"), verifying whether the predicted peptides did actually match any known gene model. And, most importantly, verifying whether the full cloned cDNA sequence matched any other protein in a different frame with respect with the frame that encoded the short peptide they inferred.

Answer: We thank the reviewer for the value assessment of our manuscript. Identifying new AMPs effective against various microbes is of great interest for developing new antimicrobial drugs to combat the growing problem of antibiotic resistance. We agree with the reviewer that several methodological approaches can be applied to identify such peptides. Sequence-based metagenomics involves directly sequencing DNA extracted from environmental samples to identify genes encoding AMPs and predict their amino acid sequences. This approach has been used to identify many new AMPs from various sources, including bacteria, fungi, and plants. Functional metagenomics involves cloning and expressing DNA fragments from environmental samples in a heterologous host, such as Escherichia coli. The resulting library of expressed proteins can be screened for antimicrobial activity, identifying new AMPs that sequence-based approaches may not have detected. Both sequence-based and functional metagenomics have advantages and disadvantages. Sequence-based approaches can identify many potential AMPs quickly and efficiently, but predicting the function of these peptides can be challenging. Functional metagenomics can provide direct evidence of antimicrobial activity, but screening large libraries can be time-consuming and labor-intensive. We decided on functional metagenomics to directly screen for the function of the peptides and characterize their effectiveness and efficiency within biofilm assays. We included a statement on our decision for functional genomics in the revised manuscript (lines 338 ff.). We are fully aware that not all peptides are from in-frame cloned sORFs – and bioinformatics would be great. Unfortunately, we cannot perform bioinformatics analysis for all 58.000 clones. It was, however, easy to check for effects on biofilms due to well-established high throughput assays in our lab. Finally, biofilm-inhibiting peptides were sequenced and analyzed. We were able to identify the sequences within the published metazoan genomes. We did not check for their expression within published transcriptome data. We know that bioinformatics analyses were not exhausted, e.g., whether the entire cloned cDNA sequence matched any other protein in a different frame with respect to the frame that encoded the short peptide we inferred; however, this was not our intention. Overall, our study focused on identifying biologically active AMPs for biofilm inhibition of opportunistic pathogens, less on identifying ecologically relevant AMPs. Obviously, ecologically relevant AMPs are an exciting research direction, and we initially intended to identify ecologically relevant AMPs of both jellies; thus, future studies will focus on those aspects. However, the presented Federal Ministry of Education and Research-funded project aimed to reveal the enormous potential of marine habitats, particularly marine animals, for identifying novel biologically active compounds potentially helpful for developing future antimicrobials. Consequently, we toned down statements on the ecological relevance of our identified AMPs throughout the revised text and highlighted the mere identification of biologically active antimicrobials.

In detail, while the authors have demonstrated the biological activity of the isolated peptides, they have briefly discussed the possibility (at line 316) that "some of the the sORFs might be artificially generated based on the experimental design, and encoded peptides identified, do not necessarily possess an ecological antimicrobial activity in the natural system". This is an important acknowledgement, but the authors should most certainly have gone even further in the characterization of cloned sequences and, most importantly, they should have NOT assumed that inserts were necessarily in frame. As a matter of fact, some of the clones reported in table S1 are clearly not in frame, and therefore the inferred peptide sequence is not a real peptide (in its natural system). I would go even further and state that most likely NONE of the peptides identified and characterized in this study was a real peptide. A clear example is provided by clone Aa_127_8E, reported to encode a portion of a retinoid X receptor. However, the analysis of the nucleotide sequence reveals that this was an out-of-frame clone of the actin mRNA (which can be clearly spotted in frame 3). Ml_011_11H is also, quite clearly, an out-of-frame 60S ribosomal protein L13 cDNA. Ml_068_11H is an out-of-frame 40S ribosomal protein S25 cDNA.

Unfortunately, this invalidates, in my opinion, the entire work. Functionally characterizing (or even predicting the hypothetical structure) of non-existing peptides does not make much sense and, albeit some of the artificial peptides generated by random translation of short (non-existing) ORFs might indeed display significant biological activities, the fact itself that such sequences are not real is completely inconsistent the rationale of the whole study (using the two target organisms as a source of novel AMPs ).

Answer: We thank the reviewer again for his/her opinion on our study design. As stated in our first answer, we primarily aimed to identify novel AMPs of marine origin by functionally screening cDNA libraries. Here, the biological activity of the peptides was the main focus. At this point, we do not want to start a fundamental discussion about a better methodology for identifying AMPs. We initially have chosen the jellies as a potential source for novel AMPs, which they might use to defend microbes. Although we could not identify naturally active AMPs, we identified artificial AMPs with promising biofilm-inhibiting activities from which structures were predicted. We rephrased the rationale of our study within the revised version of the manuscript (see introduction and discussion).

Other comments:

Please check that scientific names are written in italics throughout the manuscript

Answer: We apologize for any mistakes. We checked the manuscript carefully and removed errors.

L62: this is generally true, but I feel like this sentence is somewhat too optimistic, as only a fraction of all marine AMPs display a significant antimicrobial activity at concentrations that would make their biomedical application feasible, without showing at the same time no significant cytotoxicity. The development of AMP-based drugs for human use still has a long way to go.

Answer: We thank the reviewer for the valuable comment. We agree with the reviewer that only a certain proportion of (marine) AMPs display effective antimicrobial activity in concentrations relevant to an application. It is crucial to exclude cytotoxic effects on animals and humans. We agree that the development of AMP-based drugs is in its infancy. The revised manuscript includes hints on necessary AMP research for their potential application (lines 74 ff.) The application of our identified AMPs is wishful thinking, and tests on their cytotoxicity on human cells were not within the focus of our study. Nevertheless, negative effects on the targeted biofilm-forming bacteria have been excluded (Fig. 2).

L62: before the last paragraph of the introduction, I would suggest the authors to add a small additional paragraph to briefly summarize currently available information about ctenophore and cnidarian AMPs (i.e. no information whatsoever for the former, to the best of my knowledge, vs a few preliminary indications –including aurelin, among the others- for the latter. This would help the reader to understand that these organisms have been definitely under-exploited so far in terms of AMP research.

Answer: We thank the reviewer for this helpful comment. We added the suggested paragraph on knowledge of Ctenophore and Cnidarian AMPs to support our statement that those organisms have been underexplored (lines 57 ff.).

 Materials and methods:

One of the drawbacks of this approach is that some potentially interesting AMP sequences might have been missed due to the fact that they have a significant activity against E. coli, preventing the possibility to obtain viable bacteria expressing them. This should be somehow discussed at a later point in the manuscript.

Answer: We thank the reviewer for the valuable comment. The reviewer is correct. We potentially missed AMPs active against the K12 E. coli lab strain because of the heterologous expression of putative toxic AMPs. However, we never claimed that all jelly AMPs were identified by functional screening. We now included this aspect in the discussion section (lines 354 ff.).

The bottom part of figure 3 appears to have been cut

Answer: We apologize for the mistake. We provide the complete figure in the revised version.

Reviewer 3 Report

I have read the article entitled "Antimicrobial peptides originating from expression libraries of Aurelia aurita and Mnemiopsis leidyi prevent biofilm formation of opportunistic pathogens" with great interest and I think it is suited for a publication in the Microorganisms, Special Issue “Feature Papers in Microbial Biofilms”. The Authors have demonstrated the structural characteristics and physicochemical properties of the marine invertebrate-derived antimicrobial peptides. The peptide BiP_Aa_5 showed the strongest biofilm-preventing effects against tested microorganisms. I feel the manuscript brings some new information to the scientific community. It is well-written, clearly exposed and well structured. I am happy that references are appropriate, that tables and figures are well presented, and I can suggest that manuscript can be published in present form.

Author Response

Manuscript ID: microorganisms-2304942

Title: Antimicrobial peptides originating from expression libraries of Aurelia aurita and Mnemiopsis leidyi prevent biofilm formation of opportunistic pathogens

Authors: Lisa Ladewig, Leon Gloy, Daniela Langfeldt, Nicole Pinnow, Nancy Weiland-Bräuer, Ruth Anne Schmitz

Point-by-point response to Reviewer 3

We thank the reviewer for reviewing our manuscript and recommending publication in its present form. We thank the reviewer for taking our efforts into account.

Round 2

Reviewer 2 Report

While I understand the point of view of the authors, I stand by my opinion that biological/ecological should not be entirely ignored. Basically, what the authors reported is the activity of random stretches of amino acids that were encoded by short ORFs found out-of-frame in transcripts that had nothing to to with antimicrobial activity itself and that displayed an intersting biological activity just by mere chance.
By definition, this approach would have allowed the identification of similar peptides (not existing in nature) from any transcriptome, whether it was a rapeseed transcriptome, a donkey transcriptome or a jellyfish transcriptome.
Considering the fact that multiple in silico tools that allow the rational design of peptides with antimicrobial/antibiofilm activity de novo have been developed, I find the rationale of this study, unfortunately, extremely week.
